# Exploration of the Binding Site of Arachidonic Acid in gp63 of *Leishmania mexicana* and in Orthologous Proteins in Clinically Important Parasites

**DOI:** 10.3390/pathogens13090718

**Published:** 2024-08-25

**Authors:** Verónica Ivonne Hernández-Ramírez, Audifás-Salvador Matus-Meza, Norma Oviedo, Marco Antonio Magos-Castro, Carlos Osorio-Trujillo, Lizbeth Salazar-Villatoro, Luis Alejandro Constantino-Jonapa, Patricia Talamás-Rohana

**Affiliations:** 1Departamento de Infectómica y Patogénesis Molecular, CINVESTAV, Av. IPN No. 2508, Col. San Pedro Zacatenco, México City 07360, Mexico; vhernandezr@cinvestav.mx (V.I.H.-R.); clostrujillo2@yahoo.com.mx (C.O.-T.); lsalazar@cinvestav.mx (L.S.-V.); 2Department of Pharmaceutical Sciences, College of Pharmacy, University of Nebraska Medical Center, Omaha, NE 69198, USA; 3Unidad de Investigación Médica en Inmunología e Infectología, Centro Médico Nacional La Raza, IMSS, Av. Jacarandas S/N, La Raza, Azcapotzalco, México City 02990, Mexico; naoviedoa@yahoo.com.mx; 4Departamento de Genética y Biología Molecular, CINVESTAV, Av. IPN No. 2508, Col. San Pedro Zacatenco, México City 07360, Mexico; marcoantoniomagoscastro@gmail.com; 5Unidad de Investigación UNAM-INC, División de Investigación, Facultad de Medicina, UNAM, Instituto Nacional de Cardiología Ignacio Chávez, México City 14080, Mexico; biologia0712@gmail.com

**Keywords:** *Acanthamoeba castellanii*, COX-like activity, *Entamoeba* spp., *Leishmania mexicana* gp63, *Naegleria fowleri*, *Trypanosoma cruzi*, orthologous proteins to gp63

## Abstract

Recently, we published that the monoclonal antibody (D12 mAb) recognizes gp63 of *L. mexicana*, and it is responsible for COX activity. This D12 mAb exhibited cross-reactivity with *Trypanosoma cruzi*, *Entamoeba histolytica*, *Acanthamoeba castellanii*, and *Naegleria fowleri*. COX activity assays performed in these parasites suggested the potential presence of such enzymatic activity. In our investigation, we confirmed that wild-type recombinant gp63 exhibits COX-like activity, in contrast to a mutated recombinant gp63 variant. Consequently, our objective was to identify sequences orthologous to gp63 and subsequently analyze the binding of arachidonic acid (AA) to the putative active sites of these proteins. Given the absence of a crystallized structure for this protein in the Protein Data Bank (PDB), it was imperative to first obtain a three-dimensional structure by homology modeling, using leishmanolysin from *Leishmania major* (PDB ID: LML1) as a template in the Swiss model database. The results obtained through molecular docking simulations revealed the primary interactions of AA close to the Zinc atom present in the catalytic site of gp63-like molecules of several parasites, predominantly mediated by hydrogen bonds with HIS264, HIS268 and HIS334. Furthermore, COX activity was evaluated in commensal species such as *E. dispar* and during the encystment process of *E. invadens*.

## 1. Introduction

Arachidonic acid (AA) is the primary precursor of prostaglandins (PGs) [1]. The release of AA from cellular lipid stores mainly depends on phospholipase A2 [2]. This enzyme hydrolyzes membrane phospholipids, releasing arachidonic acid (AA). Prostaglandin-endoperoxide synthase (PTGS), also known as cyclooxygenases (COX), catalyzes the biosynthesis of prostaglandins and thromboxanes by reducing AA. COX-1 and COX-2, two isoforms of this enzyme, have similar protein structures that catalyze the same reaction [3]. The difference between the two enzymes is that COX-1 is constitutive; it is part of the homeostatic maintenance of various processes in the human body in most tissues. In comparison, COX-2 is inducible and expressed during inflammatory processes [4].

The COX/PG metabolic axis is one of the most characterized factors in the pathophysiology of host–parasite relationships since it influences the modulation of the immune response. However, the role of the COX/PG axis of parasitic origin during the disease has yet to be thoroughly studied. *E. histolytica* was the first pathogen in which COX-like activity was reported [5]. Alpha-actinin is the protein identified as responsible for COX-like activity in this parasite. Recent reports have documented the role of this amoebic enzyme and prostaglangin E_2_ (PGE_2_) in the disorganization of the tight junctions of the intestinal epithelium, contributing to the loss of intestinal barrier function and, thus, the occurrence of diarrhea [6]. Recently, our group reported that gp63 from *L. mexicana* (Lmxgp63) is responsible for COX activity [7]. Various molecular factors contribute to the virulence and pathogenicity of *Leishmania* spp. Among these factors, the major surface glycoprotein is highly relevant. Several names have been used interchangeably to describe this glycoprotein. The most used name, gp63, is derived from the 63 kDa glycoprotein, although isoforms with different masses have been found [8]. Leishmanolysin (EC3.4.24.36) was chosen from the IUBMB enzyme nomenclature, reflecting its protease activity. Leishmanolysin belongs to the M8 family of metalloendopeptidases. All M8 metalloproteases contain a zinc-binding HEXXH catalytic site motif [9]. Orthologous surface metallopeptidases exist in both *T. brucei* and *T. cruzi*, although their function differs from that of leishmanolysin [10,11,12]. Two leishmanolysin homologs are encoded in the *E. histolytica* genome (EhMSP-1 and EhMSP-2), but only one copy of the gene is present in the commensal organism *E. dispar* [13]. Current evidence shows that EhMSP-1 regulates adhesion, motility, destruction of the tissue culture monolayer, and phagocytosis [14].

In recent work, a monoclonal antibody (D12) was produced by immunizing mice with fractions enriched with the COX-like activity of *L. mexicana*. Characterization of the antibody showed that D12 recognizes Lmxgp63. Furthermore, immunoprecipitation assays confirmed that Lmxgp63 exhibits COX-like activity in the presence of AA. Cross-antigenicity of this antibody to orthologous gp63 molecules was demonstrated in other parasites using the D12 mAb in confocal microscopy [15]. In this work, we produced recombinant wild type and mutant gp63 proteins from *L. mexicana* and demonstrated that wild type rLmxgp63 protein possesses COX-like activity. Moreover, the presence of RNA transcripts further supports the existence of orthologous proteins in other parasite species. Structural analyses of the COX-2-like antigen revealed continuous and discontinuous epitopes for B cells that could be relevant to explain the cross-reaction of the D12 mAb with different parasites. In addition to the cross-reaction of the D12 mAb, the presence of COX-type activity was confirmed in soluble fractions of *E. histolytica*, the free-living amoebae *N. fowleri* and *A. castellanii* as well as the trypanosomatid *T. cruzi*. For this reason, in the present work, we set out to investigate the possible binding site of AA to proteins with possible COX-like activity of the parasite species.

## 2. Materials and Methods

### 2.1. Production of a Mutant without COX Activity: Mutation by Frameshift of the gp63 Gene of L. mexicana

The reading frame of gp63 was shifted in the pPROEX-gp63 vector to obtain a Lmxgp63 protein without COX activity [7]. The histidine tag is linked to the open reading frame (ORF) of the gp63 gene; the NdeI site lies between the histidine tag and the ORF of gp63. The construct, therefore, has a reading frame that is transcribed and translated. Modifications were made in the NdeI restriction site. The plasmid pPROEX-gp63 was cut by the enzyme NdeI and, after that, was purified with phenol-chloroform and precipitated with 1/10 volume 5 M NaCl and 2.5 volumes absolute ethanol. The plasmid was resuspended in 10 mL buffer 2X from the pJET cloning kit (Thermo Scientific Waltham, MA, USA), plus 7 mL H_2_O and 1 mL DNA blunting enzyme and incubated at 70 °C for 5 min, cooled on ice and mixed with 1 µL T4 ligase (5 Weiss units) and 1 µL H_2_O. Incubation took place for 30 min at room temperature. Ligations were transformed into *E. coli* DH5α, and the plasmid lacking an NdeI site (pPROEX-gp63fs+2 NdeI) was selected by restriction with each enzyme and verified by automated sequencing.

### 2.2. Induction and Purification of Recombinant Proteins from Leishmania Mexicana gp63

*E. coli* DH5α/pPROEX-gp63 (wt) and DH5α/pPROEX-gp63fs+2NdeI (mutant) strains were sampled to obtain 5 mL cultures in LB medium (Luria-Bertani) with ampicillin (100 mg/mL). The cultures were incubated overnight at 37 °C. The next day, a 1:100 dilution was prepared in fresh medium (LB/ampicillin 100 mg/mL) and incubated at 37 °C with constant shaking. IPTG at a final concentration of 1 mM was added until the cultures reached an optical density of 0.5 (OD 600 nm). Cultures were stimulated to produce recombinant proteins for 2 h at 37 °C. At the end, cultures were centrifuged at 3500 *×g* for 10 min at 4 °C, and supernatants were removed. The pellets were resuspended in 300 μL of lysis buffer (10 mM Tris-HCl, pH 7.5, 2 mM EDTA, 0.5 mg/mL Leupeptine, 3 mM N-ethylmaleimide (NEM), 0.5 mg/mL Aprotinin and 1 mM Benzamidine). The pellets were sonicated (10 cycles, 30 seconds of sonication, and 30 seconds of rest at 4 °C). The lysates were centrifuged at 12,000 *×g* for 10 min at 4 °C, containing a soluble fraction (SN) and pellet fraction (P). An aliquot of each sample was resuspended in a 4X sample buffer for analysis on SDS-PAGE 10 %. A replicate of this material was processed for Western blot analysis. The membrane was blocked and incubated with a commercially available His-Tag Antibody (dilution 1:1000) (HIS.H8, Santa Cruz Biotechnology Cat # sc-57598, Dallas, TX, USA). The membrane was incubated overnight at 4 °C. Finally, the membrane was washed extensively and after 1 h incubation at room temperature with the primary antibody, recombinant proteins (rLmxgp63 wild type or rLmxgp63fs+2NdeI mutant) were detected with a secondary anti-mouse IgG HRP antibody (1:1000, Thermo Fisher Cat # 31430, Waltham, MA, USA). Development was performed by chemiluminescence using the Protein biology kit (Thermo Scientific, Rockford, IL, USA) and Newton 7.0 equipment (Vilber, Collégien, France). Because the recombinant proteins were not detected in the supernatants, the pellets were treated with 0.5 % Triton-X100 in lysis buffer. Then, the samples were centrifuged at high speed, and the supernatants were collected; after that, the supernatants were incubated in Eppendorf tubes with Ni-coupled resin (500 mL/100 µL). The tubes were gently shaken overnight at 4 °C. The next day, they were centrifuged for 20 s at 4 °C and 1,000 *g*, and the supernatants were removed. The pellets were washed with 500 mL of wash buffer 1 (50 mM Tris-HCl, pH 7.0, 10 mM NaCl and 10 mM imidazole) and then with 500 mL of wash buffer 2 (50 mM Tris-HCl pH 7.0, 10 mM NaCl and 20 mM of imidazole). The pellets were resuspended in elution buffer (50 mM Tris-HCl pH 7, 10 mM NaCl, and 300 mM imidazole). Finally, a pool of eluates was made according to their origin, and they were dialyzed with 50 mM Tris-HCl pH 7.0, 10 mM NaCl, and 30% glycerol for one day at 4 °C, with three changes to a final volume of 2 L dialysis buffer. Samples were then stored at −70 °C until use in COX activity assays. In the case of the Western blot, an aliquot of the two eluates obtained from the affinity columns corresponding to the wild recombinant protein or the mutated recombinant protein was used.

### 2.3. Parasites

*Entamoeba invadens* (ATCC 30994) and *Entamoeba dispar* (SAW 760) cultures were performed according to ATCC (American Type Culture Collection, Manassas, VA, USA) recommendations and were donated by Dr. Bibiana Chávez Munguía. *Entamoeba histolytica* was cultured according to previously documented methods [16].

### 2.4. Obtaining Soluble Fractions from Membrane Components of Enamoeba spp.

Trophozoites (*E. histolytica*, *E. dispar*, and *E. invadens*) were collected at the end of the logarithmic growth phase by centrifugation for 10 min at 1500× *g* and 4 °C. In the case of cysts (*E. invadens*), cells were centrifugated at 3500× *g*/7 min. After that, the collected cells were then processed for 8–10 cycles for 1 min each, at full power in a 100-watt ultrasonic processor sonicator in lysis buffer (10 mM Tris-HCl pH 7.5, 2 mM EDTA, 0.5 mg/mL Leupeptin, 3 mM NEM, 0.5 mg/mL Aprotinin and 1 mM Benzamidine). Complete lysis of the parasites was demonstrated under the microscope. The total extracts were centrifuged at 10,000× *g*, the supernatants separated, and the pellets resuspended in 1% Nonidet P-40 in the same lysis buffer. The lysates were centrifuged for 30 min at 10,000× *g* and 4 °C. The resulting supernatant was separated, aliquoted, and stored at −70 °C until use. Protein concentration was determined using a BCA protein assay (Bio-Rad, Laboratories Inc. DC™ Protein Assay Kit II #5000112, Hercules, CA, USA).

### 2.5. COX Activity

The samples were analyzed according to the method described previously [7]. Briefly, from supernatants obtained from *E. coli* DH5α/pPROEX-gp63 (wt) and DH5α/pPROEX-gp63fs+2NdeI (mutant) with lysis buffer/ 0.5% Triton X-100, as well as soluble fractions obtained from trophozoites of the genus *Entamoeba* and cysts of *E. invadens*, the cyclooxygenase (COX) activity was determined using the COX activity kit (Cat. No. 907-003, Enzo Life Sciences, Inc., Farmingdale, NY, USA). The kit uses a specific chemiluminescent substrate to detect the peroxidative activity of the COX enzyme. COX activity is measured by adding AA. The reaction was read on a Fluoroskan Ascent FL (Thermo Electron Corporation, Waltham, MA, USA).

### 2.6. In Vitro Encystation and Fluorescence Microscopy

*E. invadens* trophozoites of strain IP-1 were cultured according to previous reports [16]. To induce encystation, trophozoites harvested in the logarithmic growth phase (10 × 10^6^/mL) were transferred to LG encystation medium (TYI medium diluted to 47% without glucose) plus 5% bovine serum. Trophozoites, round precysts (20–40 μm), and cysts (10–20 μm) were counted at 24, 48, and 72 h post-induction. Encystation kinetics samples were fixed with 4 % paraformaldehyde for 1 h at room temperature (RT). After three washes with PBS, cells were treated with 10 % bovine serum in PBS for 1 h at 37 °C. Cysts were incubated with 0.01% Calcofluor White M2R (Sigma Chemical, St. Louis, MO, USA) for 60 min at room temperature. This fluorescent dye has a specific affinity for polysaccharides such as chitin. Samples were washed three times with PBS, mounted with Vecta Shield (Vector Laboratories, Newark, CA, USA), and observed in a Carl Zeiss LSM 700 confocal microscope (Carl Zeiss, Jena, Germany). Other samples were treated for immunodetection of gp63. Briefly, cysts were washed three times with PBS and resuspended in 100 µL of PBS; 48 and 72 h samples were subjected to 3 freeze/thaw cycles with N_2_ to disrupt the cyst wall. The sample was then treated with 0.5% Triton for 20 min and blocked with 10% fetal bovine serum in PBS for 1 h. Samples were incubated overnight with α-gp63 antibody (CEDARLANE Laboratories Limited, Cat. No. CLP005A, Burlington, Ontario, Canada) at a 1:50 dilution at 4 °C. The cysts were washed five times with PBS and then incubated for 1 h with the secondary antibody (1:100 dilution of anti-mouse IgG, Alexa Fluor™ 488, Invitrogen, Waltham, MA, USA), and 1:500 dilution of Hoescht (Chemcruz, Cat. # 33258, Dallas, TX, USA). After five washes with PBS, the samples were mounted with Vecta Shield (Vector Laboratories, Newark, CA, USA) and observed in a Carl Zeiss LSM 700 confocal microscope.

### 2.7. Bioinformatics Analysis

#### 2.7.1. Bioinformatic Analyses of Local Alignment Search

The search for proteins homologous to the Lmxgp63 in different parasite species and the analyses of the percentage of identity between these sequences were performed using the BLAST program: https://blast.ncbi.nlm.nih.gov/Blast.cgi (accessed on 9 September 2020) [17].

#### 2.7.2. Data Analysis from RNA-Seq Repositories 

Using the Genbank ID or Uniprot ID of gp63 sequences of *Entamoeba*, *Trypanosoma*, *Leishmania*, *Naegleria*, and *Acanthamoeba*, we searched for the sequence in AmoebaDB: http://amoebadb.org/amoeba/ (accessed on 28 September 2023 and TriTrypDB: https://tritrypdb.org (accessed on 28 September 2023).

#### 2.7.3. Search in the Gene Ontology Database

The Gene Ontology Term (GOTerm) information was used to identify predicted functions of each sequence of putative gp63 proteins in each species. We also searched for transcript information for each sequence that could indicate a possible role in the life cycle or pathogenesis if available in the database.

#### 2.7.4. Multiple Alignments

Multiple alignments of the studied sequences were performed using the bioinformatic tool Uniprot: https://uniprot.org/ (accessed on 21 February 2023) [18].

#### 2.7.5. Conserved Domains

The research of conserved domains in the group of proteins similar to *L. mexicana* gp63 and human and mouse COX2 was carried out using the Pfam database: http://pfam.xfam.org/ (accessed on 15 April 2021) [19].

#### 2.7.6. Phylogenetic Tree

The phylogenetic tree was constructed using the Neighbor-Joining (NJ) method with a bootstrap of 1000 in MEGAX: https://www.megasoftware.net/, (accessed on 17 November 2022) [20].

### 2.8. Comparative Analysis of the Three-Dimensional Structures of L. mexicana gp63 and Proteins with COX-like Activity from Different Protozoan Parasites

To obtain these structures, all of our analyses were based on the 3D structure of Leishmanolysin from *L. major* (PDB ID: 1LML) since it is the only crystallized structure of Leishmanolysin (gp63) available in the databases. Therefore, the three-dimensional structure of Lmxgp63 and the six orthologous proteins were constructed by homology modeling using *L. major* leishmanolysin (PDB ID: 1LML) as a template with the Swiss model: https://swissmodel.expasy.org/ (accessed on 1 June 2021) [21]. Then, the alignment and superimposition of the three-dimensional structures obtained were performed using the TopMatch web service: https://topmatch.services.came.sbg.ac.at/ (accessed on 10 December 2021) [22]. 

### 2.9. Comparative Analysis of the Amino Acid Sequences of the Proteins with COX-like Activity Bound to the AA Structure

Once again, the new structures were created by homology through the Swiss model server. Leishmanolysin (gp63) from *L. major* (PDB ID: 1LML) was used as a template to obtain the structures of *L. mexicana*, *T. cruzi*, *E. histolytica*, *E. dispar*, *E. invadens*, *A. castellaniii* and *N. fowleri*. The sequences were obtained from Uniprot with entry numbers Q4DTV2, C4M655, B0ERK0, A0A0A1TW87, L8GQS8, and A0A6A5C651, respectively. Since gp63 is a zinc-dependent metalloprotease, zinc was added to the catalytic site, which was identified by the three histidine residues conserved in all our studied proteins. A similar study with *L. major* was carried out by Mercado-Camarrgo et al. [23]. Energy minimization of each structure was performed using the Yasara server: http://www.yasara.org/minimizationserver.htm (accessed on 5 December 2021), and the structures were used for molecular docking. Both the targets and the ligand (AA) in pdb format were uploaded to the SwissDock server: http://www.swissdock.ch/docking (accessed on 25 August 2023) for molecular docking. The algorithm involves generating multiple binding modes, evaluating binding energies between the target and ligand using a CHARMM-based scoring function, and selecting and clustering the lowest energy. All obtained predictions were selected for the lowest energy (i.e., the most negative value, the Gibbs free energy, ΔG) and then visualized and analyzed using the UCSF chimera package v.1.15 (RBVI, San Francisco, CA, USA) [24].

### 2.10. Statistical Analysis

All tests were performed in three independent replicates. A one-way ANOVA with the Tukey Post hoc test determined the statistical differences between the groups. All analyses were carried out in Graph Pad Prism V 8.0.2 (GraphPad Software Boston, MA, USA). 

## 3. Results

### 3.1. The gp63 of L. mexicana Is Responsible for Cyclooxygenase (COX)-like Activity

#### Obtaining the gp63 Mutant

This work reports the production of wild type and frameshift mutation and purification of the recombinant gp63 proteins (rLmxgp63 and rLmxgp63fs+2NdeI) from *L. mexicana*. To obtain a mutant that does not contain COX activity, the reading frame was shifted to another open reading frame (ORF) of pPROEX-gp63 [7] (Figure 1A). The NdeI digested plasmid pPROEX-gp63 was subsequently repaired. This NdeI site is unique in the vector and localizes the histidine tag and the start codon of the gp63 gene (Figure 1A). Two insertions in the NdeI site (pPROEX-gp63fs+2NdeI) shifted the reading frame to a position of +2 fs, which was verified by sequencing (Figure 1B red box). DH5α *E. coli* transformed with the plasmid pPROEX-gp63, or pPROEX-gp63fs+2NdeI, were analyzed. Figure 1C shows the analysis by immunoblotting using the anti-histidine antibody; the arrows indicate the molecular weight of the recombinant proteins, which corresponds to the 66 kDa protein in the case of the wild type (rLmxgp63) and to 29 kDa for the mutant protein (rLmxgp63fs+2NdeI). Since both recombinant proteins were concentrated in inclusion bodies, a solubilization process using Triton X-100 was necessary. The solubilized material was processed on a Ni-coupled resin, and the eluates were analyzed using 15 % SDS-PAGE. Figure 1D shows the protein profile of the recombinant proteins and the detection of the proteins using the anti-histidine antibody.

Finally, the COX-like activity in the mutant and wild-type proteins was analyzed. Figure 1E shows the activity detected in the Triton X-100 solubilized fraction of bacteria transformed with the mutant plasmid DH5α/pPROEX-gp63fs+2NdeI and with the wild-type plasmid pROEX-gp63, induced or not with IPTG. The soluble fraction of the empty vector was included as a negative control (pPROEX-1). The results obtained under these conditions showed that COX activity is present only in the soluble fraction of the wild-type protein induced with IPTG (Figure 1E). In comparison, the Triton X-100 solubilized fractions obtained from the mutant, non-IPTG controls, and the empty vector showed no enzyme activity. Recombinant proteins, mutant and wild type, were then purified to analyze COX activity further. Figure 1F shows that only rLmxgp63 and not rLmxgp63pPROEXfs+2NdeI, presents COX-like activity. Therefore, thanks to the production and purification of wild-type and mutated recombinant proteins, it was possible to confirm that indeed the gp63 protein is responsible for the cyclooxygenase enzymatic activity.

### 3.2. Trypanosoma cruzi, Entamoeba histolytica, Entamoeba dispar, Entamoeba invadens, Acanthamoeba castellanii and Naegleria fowleri Contain Proteins Orthologous to gp63 of L. mexicana

#### Bioinformatic Analysis for the Identification of Orthologous Proteins

We have already mentioned in previous work that several parasites exhibit COX activity [15]. As the monoclonal antibody D12 is directed against Lmxgp63 and cross-reacts with other proteins from different species, the presence of conserved discontinuous epitopes in each of the gp63 analyzed in this study (*T. cruzi*, *E. histolytica*, *A. castellanii*, and *N. fowleri*) was established [15]. Lmxgp63 is a zinc-dependent metalloprotease and is considered a molecular factor contributing to the virulence and pathogenesis of *Leishmania* parasites. This protease is also called MSP, PSP, or leishmanolysin [25]. Therefore, the next approach was to perform a bioinformatic analysis to look for the presence of orthologues proteins to Lmxgp63 in the different parasites included in this study. To carry out a preliminary search for similarities between proteins from various parasites with Lmxgp63, a local alignment bioinformatic analysis (BLAST) was performed. Table 1 shows the identified orthologous proteins to gp63 in the different parasites. Two other species of the genus *Entamoeba*, *E. dispar*, which has reduced virulence properties, and *E. invadens*, since it is the only species that can form cysts *in vitro*, were also included. The identity percentage of the *L. mexicana* gp63 and the different sequences varies between 37.86% and 25.81%. The highest percentage of identity was found for *T. cruzi* (XP_817808.1) and the lowest percentage for *E. dispar* (XP001740726.1). In the case of *E. invadens*, the protein is considered a hypothetical protein; those of *T. cruzi*, *E. histolytica* and *A. castellanii* are putative gp63 and putative leishmanolysin, respectively. In the case of *E. dispar* and *N. fowleri*, one uncharacterized protein and one unassigned protein product were identified. To complete the BLAST analysis, a reverse analysis was included, so that the E values would be expected to be identical. The result of this analysis showed that in fact they are orthologous proteins. This analysis was included in Appendix A.

Hypothetical proteins are proteins predicted from nucleic acid sequences and whose existence has not been proven. In the case of identified putative proteins in the genome, there is no evidence of the function of these proteins. In both cases it was decided to perform a BLAST transcript analysis of these proteins in RNA-seq groups of the different parasites analyzed. Figure 2 shows that *A. castellanii*, presents very low levels of mRNA. In contrast to the rest of the analyses, all the parasites presented high levels of mRNA. Previously, we reported that the D12 mAb recognizes epitopes on *A. castellanii* and *N. fowlleri* trophozoites, and this antibody has been shown to recognize discontinuous epitopes from *L. mexicana* gp63 [15]. Thus, based on BLAST and RNA seq, the evidence for gp63 protein in the studied organisms is convincing. 

**Figure 2 pathogens-13-00718-f002:**
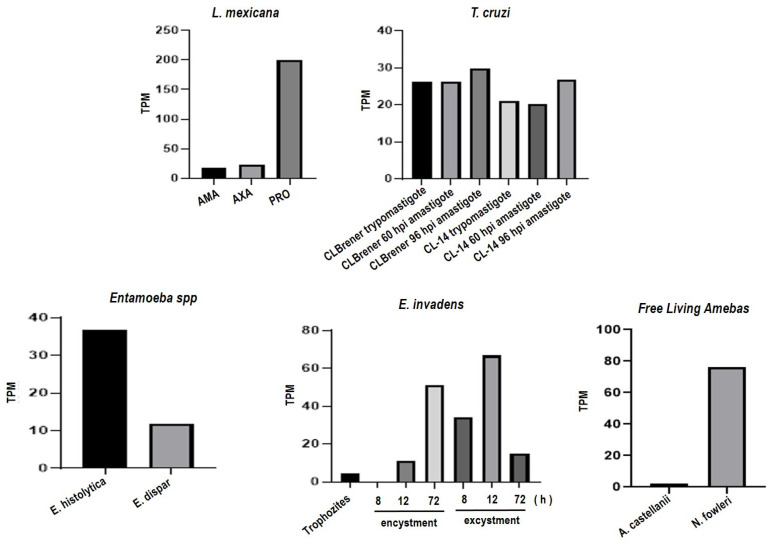
Analysis of RNA-seq data repositories from various parasites including *L. mexicana *(AMA = intracellular amastigotes; AXA = axenic amastigotes; PRO = promastigotes), *T. cruzi*, *E. histolytica*, *E. dispar*, *E. invadens*, *A. castellanii*, and *N. fowleri*. The figure illustrates the presence of transcripts per million (TPM) of mRNA corresponding to the gp63 protein in *L. mexicana* and orthologous proteins in the analyzed parasites. The TPM value indicates a relative expression level across the samples. Data for this analysis were from AmoebaDB: http://amoebadb.org/amoeba/ (accessed on 28 September 2023 and TriTrypDB: https://tritrypdb.org (accessed on 28 September 2023). Therefore, to establish a possible functional relationship between orthologous proteins with gp63, a multiple alignment was performed. Figure 3A shows the alignment of the protein sequences of each parasite. The catalytic site region is indicated on a purple background, and the HExxHAxGF motif, which is conserved across the seven proteins analyzed, can be seen. The sequences of the candidate proteins were examined using the Pfam database to determine the presence of functional domains like the *L. mexicana* gp63 (Figure 3B). The results confirmed that all analyzed sequences share the same domain of the M8 peptidase family. This domain was reported to be present in *L. mexicana* gp63 [25]. This enzyme is found in eukaryotes, including *Leishmania* and other protozoan parasites. A domain of the Transcription Factor Immunoglobin (TIG) family, called IPT/TIG, was identified in the *E. histolytica* protein. This domain is characterized by a fold like that of immunoglobulin and is found in tyrosine kinase receptors such as Met and Ron receptors. In addition, this domain is also present in transcription factors involved in DNA binding [26] (Figure 3B).

A phylogenetic tree was constructed to infer the evolutionary history of the studied proteins, using the gp63 protein from *L. mexicana* and the orthologous proteins of the six protozoa species. Figure 4 shows an ancestral relationship between the sequences of *L. mexicana* and the proteins orthologous to gp63 of *T. cruzi;* thus, these two taxa are the most closely related. The next closest related taxon is *A. castellanii*, which shares an internal node with the mentioned taxa. Concerning the genus *Entamoeba*, the COX-like activity sequence of *E. histolytica* is preferentially linked to a node within *A. castellanii*. Finally, the *N. fowleri* sequence is phylogenetically related to an internal node of *A. castellanii*. 

### 3.3. Orthologous Proteins Have Characteristics of Metalloproteinases 

#### Ontology of Genes Orthologous to gp63 of *L. mexicana*


In addition, the sequences and predicted functions of proteins orthologous to *Leishmania* gp63 were analyzed using AmoebaDB for Amoeba species or TriTrypDB for trypanosomatids. Each sequence used in this work was identified in the database. The primary molecular function identified for each sequence of *E. histolytica*, *T. cruzi*, *L. mexicana*, *E. dispar*, *E invadens*, *A. castellanii*, and *N. fowleri* was the metalloendopeptidase activity (GOTerm: GO: 0004222). The GOTERM for proteolysis (GO: 0006508) and cell adhesion (GO: 0007155) were also proposed for each sequence. A characteristic feature of metalloproteinase proteins is the need to bind zinc for their enzymatic activity. This function is presented as metal ion binding (GOTERM: GO: 0046872) in both *L. mexicana* and *T. cruzi;* only *E. histolytica* of the amoeba species has the GOTERM for zinc ion binding (GO: 0008270). The GOTERMs assigned for each sequence are listed in Table 2.

These results suggest that each sequence is a gp63-like protein, although only three of them have gp63 in their name entry: *E. histolytica*, *T. cruzi*, and *L. mexicana*. The others include hypothetical proteins with this name (*E. dispar* and *E. invadens*), a putative leishmanolysin (*A. castellanii*), and a metalloendopeptidase zinc ion binding protein (*N. fowleri*).

### 3.4. The gp63 of L. mexicana, and the Orthologous Proteins Present in T. cruzi, Entamoeba histolytica, Entamoeba dispar, E. invadens, A. castellanii, and N. fowleri, Have a Probable Binding Site for Arachidonic Acid

#### 3.4.1. Conformational Analysis of gp63

The 3D analysis performed with TopMach-web is a computational tool used to analyze protein structure alignments [22]. Since the crystallized structure of gp63 from *L. mexicana* has yet to be reported, the 3D structure was modeled using the leishmanolysin of *L. major* (PDB ID: 1LML) as a template using the Swiss model. To proceed with the identity analyses between leishmanolysin from *L. major* and *L. mexicana*, the structural similarity between these two molecules was determined. Appendix A shows the structural similarity (3D) as well as the similarity in amino acid composition of the proteins, leishmanolysin and gp63 from *L. major* and *L. mexicana*, respectively. Panel 1Aa shows the overlay corresponding to the 3D molecules; structurally equivalent parts of the proteins appear in red, and in green the structures that are not shared. Panel Ab shows the BLAST analysis, and both proteins presented 81.06% of identity and with a coverage of 97%. Panel 1B shows a detailed comparison, both structural (1, 2, and 3, upper panels) and sequences (1′, 2′, and 3′, lower panels), between these two proteins. Similar structures that consist of highly conserved amino acid sequences are shown in red that correspond to the region between residues 109 and 209 (Figure 1B 1,1′); the region between residues 419 and 489 (1B 2,2′); and the region between residues 509 and 569 (1B 3,3´). In these last two regions, only the partial sequence (1′, 2′, 3′) of what was observed in the 3D structure is shown.

This result is consistent with data obtained from the analysis of the crystallized structure of PDB ID: 1LML from *L. major* with the leishmanolysin sequence (UniProt ID A0A088RJX7) from *L. panamensis* [23]. Therefore, leishmanolysin of *L. mexicana* is orthologous to gp63 from *L. major*. Then, the proteins of all parasites modeled in the Swiss model were included in this analysis. 

Having established the identity between Leishmanolysin of *L. major* and gp63 of *L. mexicana*, it was decided that the 3D structure of the orthologous proteins should be analyzed using Leishmanolysin of *L. major* (PDB ID: 1LML) as a template.

Figure 5A shows the 3D structural models of *L. major* Leishmanolysin (left panel) and *L. mexicana* gp63 (middle panel) and their overlay (match). As shown in the matchboxes, the structures observed in orange and red correspond to the structurally aligned residues, with the structure shown in orange representing the query protein and those in red representing the target protein. Structures in green or blue correspond to unaligned residues of the query and target proteins, respectively. A high similarity was also observed when the same analysis was performed, now between *L. major* leishmanolysin and orthologous proteins of *T. cruzi* (Figure 5B) of *Entamoeba* sp. (Figure 5C), and free-living amoebae (Figure 5D). 

The values obtained by the analysis with TopMatch-web show that the analyzed structural models are very similar regarding the structure of *L. major* gp63 since the value of aligned residues (Length) and the measurement of aligned residues (Score) were very similar (Table 3). The root mean square deviation (RMS), which determines the distance of the overlapping residues of aligned structures, was between 0.44 and 0.79. Therefore, these results suggest that the query and target sequences are similar. For the *L. mexicana* gp63 protein, superimposing the three-dimensional structure with the structure of the 1LML protein gives an RMS value of 0.07, as shown in Table 3, since they belong to the same genus. Furthermore, their amino acid sequence alignment showed 81% identity (see Appendix A; and Table 3, value SI %).

#### 3.4.2. Molecular Docking Analysis

After confirming the similarity among all the analyzed orthologues proteins with *L. major* leishmanolysin, a molecular docking analysis was performed to analyze the likely binding site for the AA in these structures. Figure 6A shows in the left panel, the 3D structure of leishmanolysin from *L. major* at the bottom and its catalytic site (Figure 6B). The middle panel shows the result of the molecular docking of AA to the different structures of the orthologues to gp63 of *L. mexicana* of the different parasites analyzed. Applying the algorithm used by SwissDock, AA (shown in purple) was found to have a preferential affinity toward the catalytic site in most molecular targets, except for *T. cruzi* and *E. invadens*, where the affinity occurred at a non-catalytic site. Although the amino acids of the histidine triad along with the glutamic acid are present in the motif (HExxHAxGF) (Figure 3A), most likely, the presence of the other amino acid residues in the area surrounding the motif creates a conformation that changes the nature and affinity of the histidine triad catalytic site for AA. The rightmost panel (Figure 6B) shows a close-up of the interaction between the catalytic site and the position occupied by the AA in each protein that maintains their affinity. The three histidine and glutamic acid are observed together with the zinc atom. 

We can also observe that each AA (shown in color depending on the protein under study) took a different arrangement due to the high degree of freedom of the polyunsaturated hydrocarbon chain. However, the orientation of the carboxylic acid of the AA molecules is toward the zinc atom, forming hydrogen bonds to histidine. The types of interactions found in the molecular docking of AA to each of the analyzed proteins are described in detail below. Figure 7 shows the interaction of AA with the catalytic site and the interaction of the zinc atom with the carboxylic acid of AA for *L. mexicana*, *E. histolytica*, *E. dispar*, *A. castellanii*, and *N. fowleri* (highlighted images with thick border). It also depicts hydrogen bonding interactions with HIS264, HIS206, and HIS267 for *L. mexicana*, *E. histolytica*, and *E. dispar*, respectively. In the case of *T. cruzi* and *E. invadens*, the AA was in a pocket distant from the catalytic site, and only *T. cruzi* showed hydrogen bonds with ARG416 and TYR379. In both *A. castellanii* and *N. fowleri*, AA also interacted with the catalytic site. However, no hydrogen bonds with histidine residues of the catalytic site were detected with the algorithm used in this study, and the interactions they did show were predominantly hydrophobic. 

Likewise, it is important to mention that the binding energy found in both organisms was one of the most favorable (Table 4). Finally, Table 4 summarizes the main observed interactions between AA and the modeled proteins, and the hydrogen positions found. In addition to the triad of histidine residues forming the catalytic site, the presence of glutamic acid was also preserved. Interestingly, the two modeled proteins from *T. cruzi* and *E. invadens* have less preferred binding energies than those interacting with the catalytic site. 

### 3.5. The Entamoeba Genus Contains a COX-like Activity Which Is Present during the Encystment Process of E. invadens

#### 3.5.1. Determination of COX Activity from Soluble Fractions of *E. histolytica*, *E. dispar* and *E. invadens*, Using Exogenous AA and the Commercial COX Activity Kit

The presence of COX-like activity has been reported in the promastigote and amastigote forms of *L. mexicana* [7]. Therefore, the possibility of a COX-like involvement in the life cycle of parasites could be conceivable. In this context, *E. invadens* was included in this study because, under culture conditions, it is possible to maintain the two phases of the cell cycle: the cyst and the trophozoites. Therefore, COX activity was first determined with the commercial kit, using exogenous AA in the supernatants extracted from the different extracts of trophozoites of the genus *Entamoeba*. Figure 8A shows COX-like activity in *E. histolytica*, *E. dispar*, and *E. invadens*. The results obtained show that all species have COX activity. Assays were performed with normalized protein concentration, using a final 50 mg/25 mL concentration when analyzing COX activity. Therefore, we propose that *E. histolytica* exhibits the highest COX activity compared to *E. dispar* and *E. invadens* under these conditions.

#### 3.5.2. Detection of COX Activity during the Encystment of *E. invadens* Trophozoites

After confirming that *E. invadens* has COX activity, we induced encysting trophozoites to purify cysts at various points of the process. Figure 8Ba shows an aliquot of cysts analyzed with calcofluor using confocal microscopy. This compound binds to chitin, a carbohydrate located in the cyst wall (48 h after cyst induction). Figure 8Bb shows the purity of the sample by DIC using confocal microscopy. A replicate of this material was analyzed, and Figure 8C shows the presence of COX activity during encystation kinetics, demonstrating a maximum activity at 48 h post-induction. In addition, immunofluorescence analysis was performed during the encystment process of *E. invadens*, using the commercial monoclonal antibody anti-gp63 from *L. major*. Although cross reaction with this antibody occurs early after induction of encystation (24 h), Figure 8D shows the cross-reaction after 48 h after of encystment (Figure 8Da), when the highest amount of the antigen was detected. At 24 h, only a few cells were recognized by the antibody. The recognition pattern remained until 72 h (data not shown), strongly suggesting the presence of gp63 in cysts. DIC was used to confirm the purification process of the cysts (Figure 8Db).

## 4. Discussion

Eicosanoids are active lipid products often derived from arachidonic acid (AA). They are produced primarily via several enzymatic pathways, including cyclooxygenases (COX), lipoxygenases (LOX), and cytochrome P-450 epoxygenase (CYP450). Alternatively, a small proportion of eicosanoids are formed by autoxidation of AA [27]. In recent years, research has focused on the role played by the COX/PG axis in parasites during the pathogenic processes and life cycles of these organisms. Parasites have the necessary machinery to synthesize eicosanoids; however, few reports document cyclooxygenase activity in protozoa. In this context, the only description of both a COX-like activity and the role of prostaglandins in the pathogenicity process was reported in the protozoan parasite *E. histolytica* [5,6]. In the case of *Leishmania*, it was documented that it can metabolize AA to prostaglandins using cyclooxygenase (COX) activity [7]. In this context, it has just been published that other parasites could also have this activity [15].

In this work, it is now demonstrated that a gp63 mutant protein from *L. mexicana* lacking the catalytic active site of the protease is unable to process arachidonic acid, the substrate of COX activity. In addition, by bioinformatics analysis, we report that medically important parasites possess hypothetical proteins orthologous to *L. mexicana* gp63. This is the case for *E. invadens*, whereas a putative orthologous protein was identified in *E. histolytica*, *T. cruzi*, and *A. castellanii*. In contrast, in *E. dispar* and *N. fowleri*, an uncharacterized protein and an unnamed protein product were identified, respectively. *T. cruzi* Tcgp63-I and *E. histoltica* EhMSP-2 were identified as orthologous proteins to *L. mexicana* gp63. *T. cruzi* was shown to possess ten or more gp63 or gp63-like genes, as in most *Leishmania* spp. Two of these groups, Tcgp63-I and -II, are present as high-copy number genes. Tcgp63-I encodes surface proteins attached to the membrane through a GPI anchor, with a molecular weight of ~78 kDa, which are differentially expressed during the life cycle of the parasite. In contrast, the Tcgp63-II group is barely expressed. Both Tcgp63 and gp63 from *Leishmania* spp have the consensus sequence for the zinc-binding site, a region associated with metalloprotease activity (Figure 3A). Likewise, the most critical residues for the catalytic activity, both His and Glu in the HEXXH motif, are completely conserved in the Tcgp63-I and Tcgp63 II groups [11,28,29]. To identify which of the proteins (Tcgp63-I and Tcgp63 II) was the orthologous protein, blast analysis was performed using Tcgp63-I and Tcgp63-II as target sequences and the *L. mexicana* gp63 as the query sequence. The result showed that the identified protein (XP_817808.1, Table 1) had 76.94% identity with the protein belonging to “a” members of the Tcgp63-I group. (see Appendix A). A possible involvement of the orthologous enzyme gp63 in the life cycle of this trypanosomatid should be investigated in future work, with particular emphasis on the AA binding site, which was theoretically established in *T. cruzi* by hydrogen bonding with TYR379 and ARG416 (Table 4). This analysis may be essential to correlate COX activity with the life cycle of this parasite. In the case of *E. histolytica*, the genome has been documented to contain two homologs of the leishmanolysin metalloprotease gene, *E. histolytica* MSP-1 and MSP-2. The nomenclature of this leishmanolysin corresponds to EhMSP-1 and EhMSP-2 [14] (NCBI GeneID: numbers 3409717 and 3406949, respectively [14]); while the commensal amoeba *E. dispar* lost EhMSP-1. By searching for the products of these genes and comparing them with the orthologous sequence identified in the study (XP_652632.1, Table 1), we confirm that the orthologous protein corresponds to EhMSP-2. Recent studies have shown that EhMSP-1 is involved in the regulation of amoeba adhesion, with additional effects on cell motility, disruption of cell monolayer, and phagocytosis [14]. More recently, the underlying mechanisms of adhesion and altered motility in EhMSP-1 deficient trophozoites have been shown to be the basis for identifying critical kinases and phosphatases for the control of amoebic invasiveness [27]. However, in the case of EhMSP-2, studies have not addressed a role of EhMSP-2 during these processes. Previous work by our group showed the presence in soluble fractions of *E. histolytica* trophozoites of proteins antigenically related to mouse COX, identified in Western blot and in immunofluorescence assays, using a commercial anti-mouse COX antibody. Although the existence of α-actinin-associated cyclooxygenase activity in *E. histolytica* has been previously reported [5], it is very likely that this parasite possesses more than one protein with this activity, since in this work we are demonstrating that EhMSP2 can bind arachidonic acid. Therefore, *E. histolytica* could potentially possess various COX-like activities. Regarding the hypothetical orthologous proteins identified in the *E. dispar*, *E. invadens*, and *N. fowleri* parasites, the analyses in predictive functional platforms, as well as the analyses in mRNA repositories, revealed that the identified proteins correspond to surface proteins with metalloendopeptidase activity, where mRNA levels were detected in the case of *E. invadens* and *N. fowleri*. Furthermore, in the case of the encystation process of *E. invadens*, in the immunofluorescence analyses, the presence of crossed antigenicity was determined using the commercial anti-gp63 antibody of *L. major*. Furthermore, biochemical analyses with exogenous AA (20 mM) revealed the presence of COX activity in *E. invadens* and *E. dispar* (Figure 8A). When encystation kinetics were performed, a maximum peak of activity was observed 48 h after the onset of encystation. In the case of *E. histolytica*, the presence of this activity was confirmed, as previously documented [15]. Recently the genome of *N. fowleri* was purified using Oxford Nanopore Technology (ONT). This method assembled and polished the long reads, enabling the conservation of a high-quality genome [28]. In this context, the protein sequence, identified in this study as a protein orthologous (XP_044566011.1) to *L. mexicana* gp63, was preserved In the case of this gp63-type protease, it would be necessary to define whether it participates during the invasion process, in the encystation process, or both functions. We are currently cloning the gene corresponding to the unnamed protein product (XP_044566011.1, Table 1) to deepen the knowledge of this orthologous protein in the biology of this free-living amoebaTherefore, we can speculate that the parasitic protozoan species *E. dispar*, *E. invadens*, and *N. fowleri* may express gp63-like proteins, with the biological function of these proteins, either during the host-parasite interface or during the life cycle of the protozoan, remaining to be analyzed in the future.. Concerning *A. castellanii*, the putative orthologue protein leishmanolysin was identified with accession number XP_004337275 (Table 1). In this context, Gene Ontology Term (GOTerm) analysis suggests that this protein has the function of a zinc-binding metalloendopeptidase. However, analysis of RNA repositories has not yet revealed the presence of mRNA for this putative leishmanolysin protein. In this context, we recently reported both the presence of COX activity in soluble fractions of *A. castellanii* trophozoites, and a cross-reaction with the D12 monoclonal antibody that recognizes *L. mexicana* gp63 [15]. In addition to this evidence, it is important to highlight that when reviewing both the prediction function data and the multiple alignment analyses, it was confirmed that the analyzed proteins have zinc-binding sites, as is the case with Tcgp63 protein II (XP_817808.1, Table 1) and EhMSP-2 (XP_652632.1, Table 1), as well as in the case of the putative leishmanolysin protein from *A. castellanii*. The potential binding site for this ion was also identified in the hypothetical orthologous protein present in *E. invadens*; uncharacterized protein in *E. dispar*; and unnamed protein product *N. fowleri*. On the other hand, the domain analysis and the predictive functional analysis revealed that all examined proteins are classified into the M8 peptidase family. This implication needs to be validated experimentally, focusing primarily on the hypothetical, unnamed and uncharacterized orthologous proteins, because no information is available. In the case of the AA binding site, the species infecting humans were found to have similar binding sites (HIS206 and HIS267 and GLU207 for *E. histolytica* and *E. dispar*). In the case of *E. invadens*, the binding site was established through hydrophobic interactions, suggesting the relevance of the environment in which these amoebae parasitize. In the genus *Entamoeba*’s case, the results could suggest an involvement of the orthologue proteins in the life cycle of these parasites, especially in the encystation process (Figure 8D,E). In this regard, a previous report published by Siddiqui et al., [29] the non-steroidal anti-inflammatory drug (NSAID), sodium diclofenac, which targets the activity of COX, was shown to affect the growth but not the viability of *A. castellanii*. It is relevant to notice that NSAIDs (sodium diclofenac and indomethacin) abolished encystization in *A. castellanii*. Therefore, the authors propose that cyclooxygenases and prostaglandins of parasitic origin play an essential role in the biology of *Acanthamoeba*. However, COX activity has been reported for α-actinin of *E. histolytica*, whose sequence lacks domains and residues necessary for COX activity. Among these residues, we can mention the presence, in the COX-like activity of *L. mexicana*, of the proton-accepting histidine at position 193, the metal-binding histidine at position 374, and tyrosine at position 371 within the active site [7]. In the case of the gp63 sequence, the multiple alignment allowed us to identify the presence of glutamic acid and three metal-binding histidine (zinc) residues that could interact with tyrosine residues at positions 353 or 354 that are close to the active site (Figure 3A *L. mexicana*). gp63 is a zinc-dependent metallopeptidase anchored to the plasma membrane by a glycosylphosphatidylinositol (GPI) tail, although hydrophilic and secreted isoforms have also been described. It belongs to the enzyme class EC 3.4.24.36 (MA clan, M8 endopeptidase family) and shares several similarities with mammalian matrix metallopeptidases [30]. In *Leishmania* spp., the functions described for this protein include various cellular processes [30]. Recent findings [7,15] have shown that gp63 from *L mexicana* can metabolize AA, indicating the role of gp63 as a multifunctional protein. In *E. dispar*, *E. invadens*, and *N. fowleri*, the description of proteins orthologous to gp63 of *L. mexicana* is new, but not in *E. histolytica* and *T. cruzi* since these proteins have already been identified as leishmanolysins; however, the role of these proteins in the metabolism of AA has not yet been described. A putative leishmanolysin protein has already been described in *A. castellanii* [31] and there is also a report indicating the possible presence of COX activity [29]. Considering that the genes encoding gp63 are organized in tandem, it will be necessary for future research to show whether COX-like activity would arise from all these encoded proteins. Recently published work compared active sites residues and protein sequences from gp63 with COX activity and prostaglandin F2α synthase (PGFS) in *Leishmania* species and showed an alignment and phylogenetic analysis of GP63 and PGFS indicated notable similarity and homology between the Old and New World *Leishmania* spp. [32].

Additional results revealed that PGFS enzymes increased during parasitic differentiation in various *Leishmania* species; and that the gp63 enzyme with COX-like activity reduces during *L. braziliensis* and *L. infantum* promastigotes differentiation, but no differences were observed in *L. amazonensis* [32]. Finally, the results obtained with the molecular docking analyses suggest that proteins orthologous to gp63 could be targets for studies to develop drugs whose central purpose would be to control and/or interrupt the transmission cycle in the infection of the various parasites analyzed that are of clinical importance in this study.

## 5. Conclusions

COX activity was determined in the presence of exogenous AA in extracts of *E. dispar* trophozoites and in both *E. invadens* trophozoites and cysts. Likewise, after *in silico* analyses performed on gp63-like protein sequences in various parasites of clinical importance, such as *T. cruzi*, *E. histolytica*, *E. dispar*, *E. invadens*, *A. castellanii* and *N. fowleri*, these proteins were found to be orthologous to gp63 of *L. mexicana*. Therefore, we suspect these are likely the proteins responsible for COX-like activity in these parasites due to the presence of a theoretical sequence pointing to a probable AA union site. Performing site-directed mutations in these theoretical sequences will be vital to validating our hypothesis.

## Figures and Tables

**Figure 1 pathogens-13-00718-f001:**
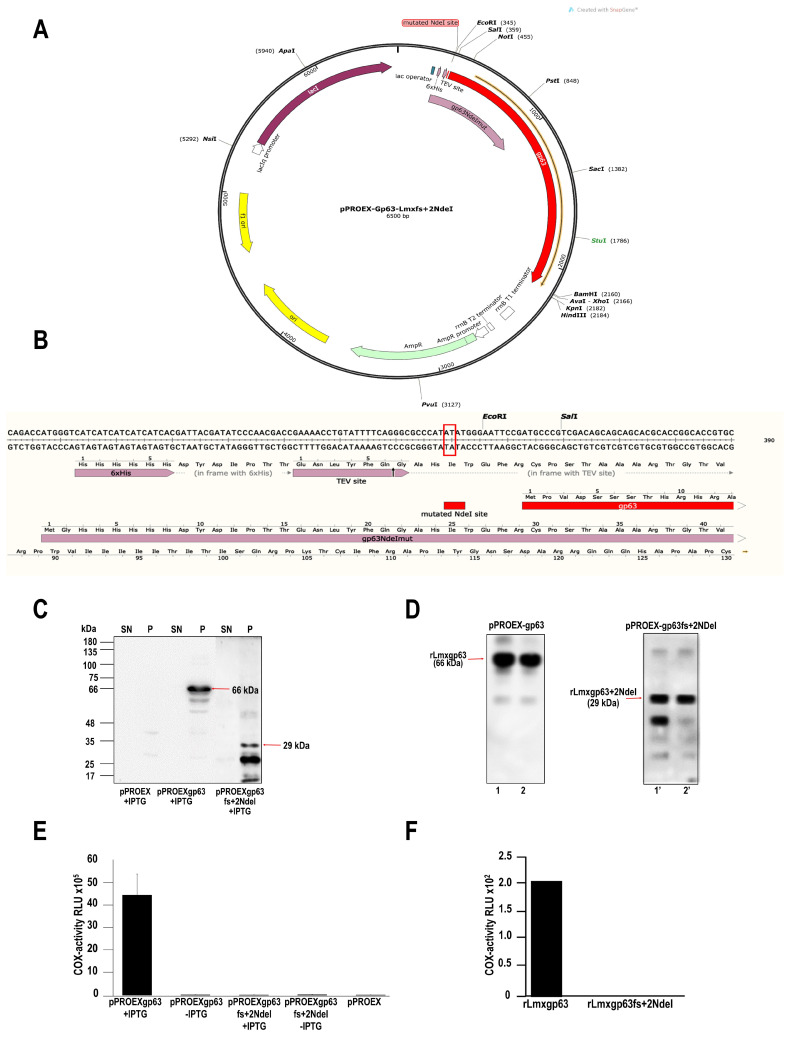
Role of the *L. mexicana* gp63 protein in COX-like activity. (**A**) The plasmid map highlights a mutated *Nde*I site in the *L. mexicana* His-tag-gp63 protein, with a new peptide indicated by a pink arrow produced by a different reading frame. (**B**) Changes in the NdeI site’s reading frame in amino acids are shown for both wild type and mutant proteins. The red box signifies the insertion of two nucleotides, resulting in a 262 amino acid product with a molecular weight of 29 kDa. (**C**) Western blot analysis of proteins from *E. coli* DH5α transformed with wild-type (pPROEX-gp63) and mutant (pPROEX-gp63fs+2NDeI) plasmids on a 10% SDS-PAGE gel. Plasmid pPROEX served as a control. Recombinant gp63 proteins were detected using a commercial anti-tag histidine antibody (1:1000) in the presence of IPTG. Supernatant = SN; pellet = P. (**D**) Recombinant protein purification from Triton-X100 solubilized pellets of rpLmxPROEX-gp63 and rpLmxPROEX-gp63fs+2NDeI transformed bacteria, followed by Ni2+ resin incubation and analysis via 15 % SDS-PAGE and Western blot. Lanes 1 and 2 correspond to two eluates of the protein and wild recombinant, and in lanes 1′ and 2′, two eluates corresponding to the mutant recombinant protein are shown. (**E**) Assessment of COX-like activity in wild-type and mutant constructs’ total extracts using a COX activity kit, with the vector serving as a negative control. These experiments were carried out in triplicate conducted in three independently performed biological assays. Purified recombinant proteins were concentrated and subjected to dialysis with renaturation buffer. (**F**) COX activity detected in purified recombinant proteins, with analyses conducted in triplicate across two independent experiments.

**Figure 3 pathogens-13-00718-f003:**
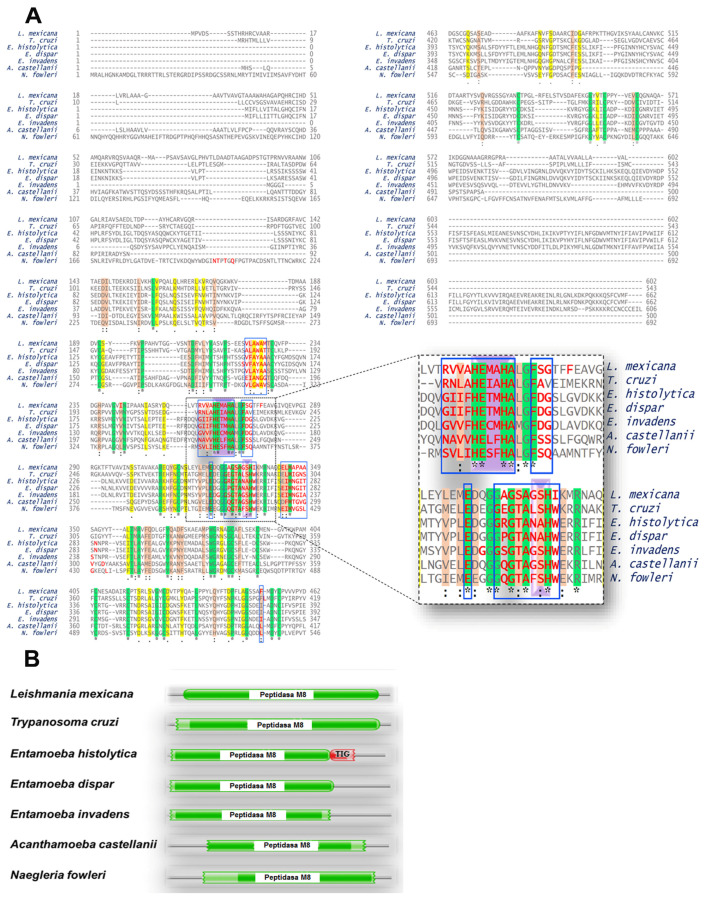
(**A**) Multiple sequence alignments of proteins like the *L. mexicana* gp63 protein (XP_003872886.1) alongside counterparts from *T. cruzi* (XP_817808.1), *E. histolytica* (XP_652632.1), *E. dispar* (XP_001740726.1), *E. invadens* (XP_004184102.1), *A. castellanii* (XP_004337275.1), and *N. fowleri* (XP_044566011.1). The alignment, generated using Uniprot online platform, highlights highly conserved or identical residues in red. The green shading indicates identical residues, while light brown and yellow denotes moderately and low conserved residues, respectively. Red letters within blue boxes represent regions within 10 Å of the zinc atom, while the purple background highlights the catalytic site. (**B**) Conserved domains in proteins homologous to gp63. All examined proteins contain the leishmanolysin domain, belonging to the M8 peptidase family, spanning specific regions: 46–570 in *L. mexicana*, 58–510 in *T. cruzi*, 27–490 in *E. histolytica*, 28–490 in *E. dispar*, 32–423 in *E. invadens*, 103–406 in *A. castellanii*, and 222–632 in *N. fowleri*.

**Figure 4 pathogens-13-00718-f004:**
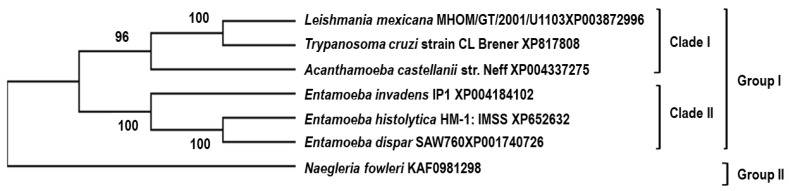
A phylogenetic tree was constructed for proteins similar to gp63 from *L. mexicana*. The tree was generated using the complete amino acid sequences of the studied proteins via the neighbor-joining method, employing the phylogenetic inference package (PHYLIP) version 3.5c. The construction utilized the amino acid sequence of gp63 from *L. mexicana* along with orthologous sequences from various parasites, including *L. mexicana* (XP_003872886.1), *T. cruzi* (XP_817808.1), *E. histolytica* (XP_652632.1), *E. dispar* (XP_001740726.1), *E. invadens* (XP_004184102.1), *A. castellanii* (XP_004337275.1), and *N. fowleri* (XP_044566011.1).

**Figure 5 pathogens-13-00718-f005:**
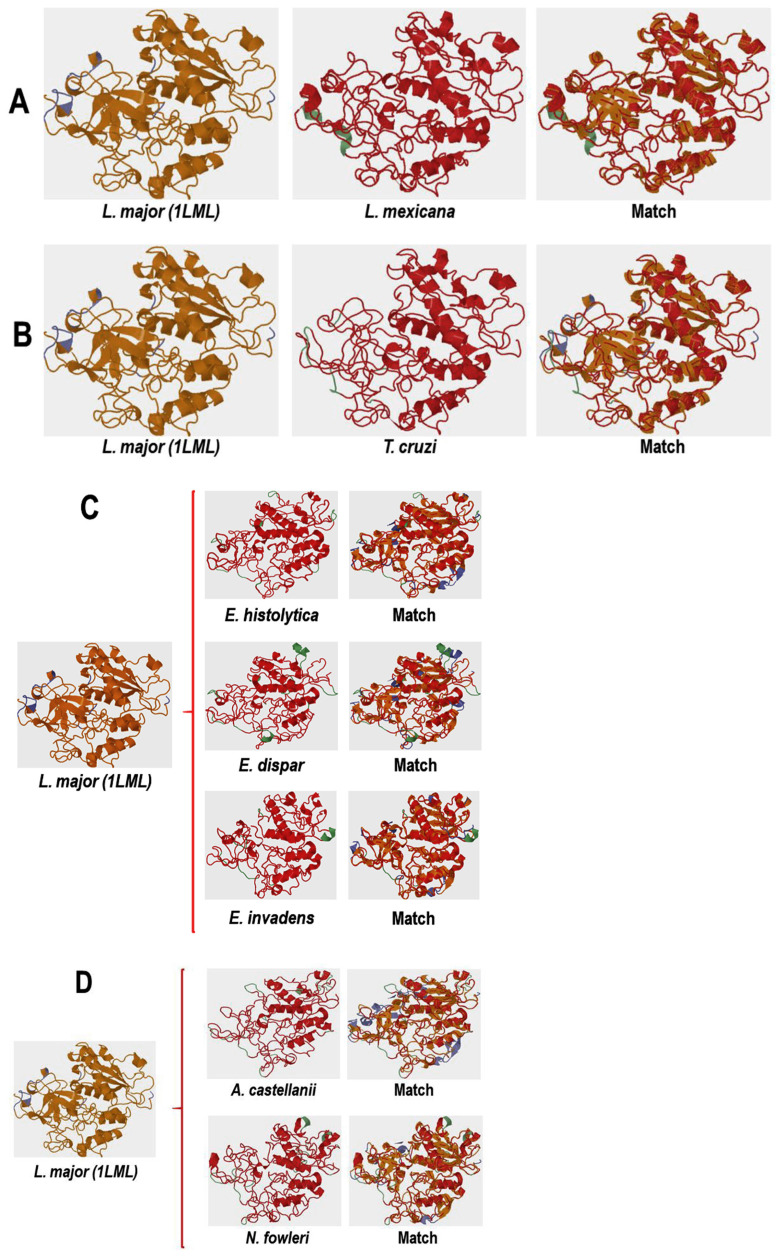
Comparing the three-dimensional structures of proteins orthologous to *L. mexicana* gp63. The 3D structures of the analyzed proteins were modeled using the Swiss model. Subsequently, the overlap analysis of these structures was conducted using TopMatch-web. In the visual representation, structurally aligned sequences are highlighted in orange and red to indicate a match, while dissimilar structures are depicted in green or blue. (**A**) Comparative analysis between *L. major* (PDB ID: LML1) and *L. mexicana* (XP_003872886.1) is illustrated. (**B**) Comparison between the leishmanolysin (PDB ID: LML1) of *L. major* and protein XP_817808.1 of *T. cruzi* is shown. (**C**) Sequences from the genus *Entamoeba* (*E. histolytica* XP_652632.1, *E. dispar* XP_001740726, and *E. invadens* XP_004184102.1) are compared with the leishmanolysin (PDB ID: LML1) from *L. major*. (**D**) Comparative analysis with free-living amoebae, *A. castellanii* and *N. fowleri* (XP_004337275.1, XP_044566011.1), respectively, is presented.

**Figure 6 pathogens-13-00718-f006:**
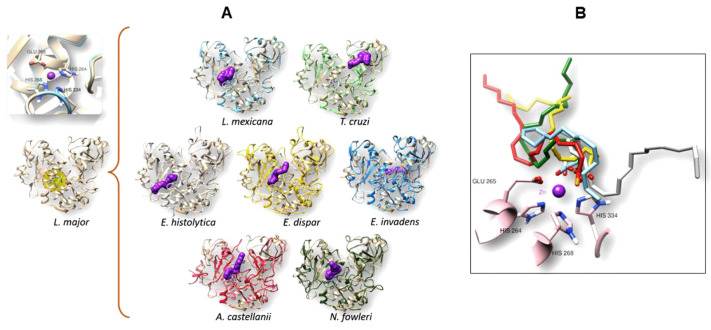
Molecular docking between arachidonic acid (AA) and each studied target. (**A**) Displays overlapping structures of *L. major* with each model obtained by homology modeling. The surface of the ligand is highlighted in purple. (**B**) Provides an expanded view of the catalytic site of *L. mexicana*, showcasing the superposition of each obtained ligand. The ligands showing affinity for the catalytic site of the molecular docking study are represented as follows: light blue for *L. mexicana*, light green for *T. cruzi*, gray for *E. histolytica*, yellow for *E. dispar*, blue for *E. invadens*, red for *A. castellanii*, and dark green for *N. fowleri*.

**Figure 7 pathogens-13-00718-f007:**
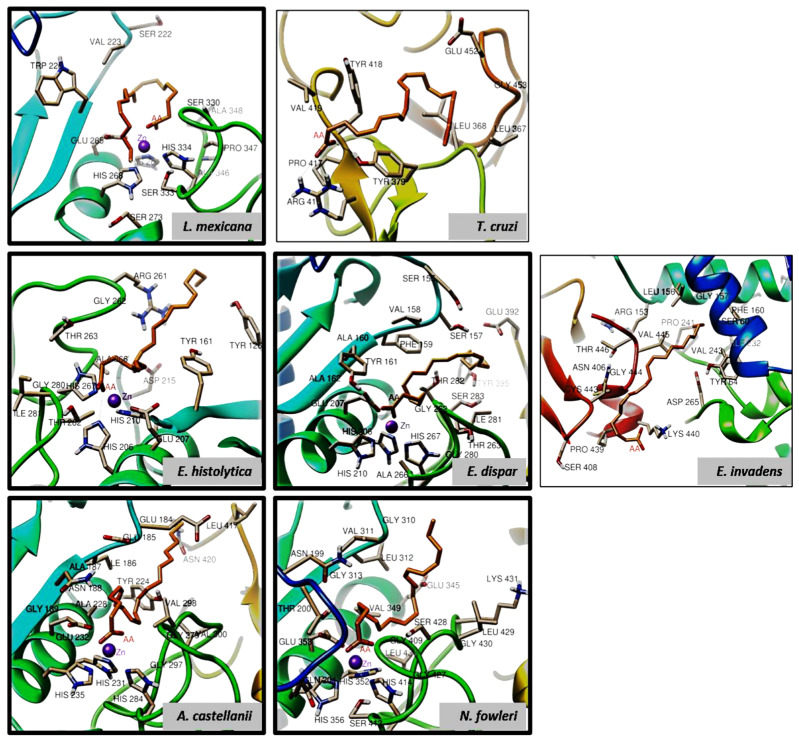
The binding modes of arachidonic acid (AA) with targets of each parasite are as follows: *L. mexicana* involves hydrogen bonds with HIS264; *T. cruzi* shows hydrogen bonds with TYR379 and ARG416; *E. histolytica* forms hydrogen bonds with HIS206; and *E. dispar* establishes hydrogen bonds with HIS267 and GLU207. For *A. castellanii* and *N. fowleri*, the binding between the gp63-like proteins and AA occurs through hydrophobic bonds. The interaction of the zinc atom with the carboxylic acid of AA is shown in the images with a thick border.

**Figure 8 pathogens-13-00718-f008:**
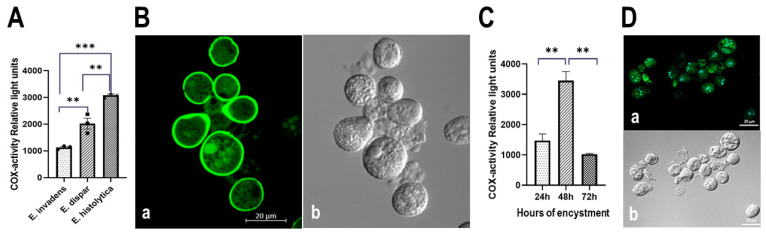
*E. histolytica*, *E. dispar*, and *E. invadens* exhibit COX-like activity. (**A**) COX activity detection in the genus *Entamoeba* involved using a commercial kit and exogenous AA in soluble fractions (50 µg protein/25 µL). (**B**–**D**) Presence of COX-type activity during the encystment process: n (**Ba**), cyst presence was confirmed by calcofluor staining, with cyst purity validated through DIC microscopy analysis (**Bb**). This sample represents the 48-hour encystment process. COX-type activity determination during encystment utilized the commercial kit and exogenous AA, with trophozoite and cyst extracts’ soluble fractions tested using exogenous AA at a final concentration of 20 µM (**C**). In (**Da**), gp63 presence in 48-hour-induced cysts was observed using anti-*Leishmania major* gp63 monoclonal antibody (CEDARLANE Laboratories Limited, Cat. No. CLP005A, Burlington, Ontario, CA), with the reaction developed through incubation with anti-mouse IgG coupled to FITC. Sample purity was analyzed through DIC microscopy (**Db**). Three independent biological replicates were conducted in triplicate *p* ≤ 0.01 **, *p* ≤ 0.001 ***.

**Table 1 pathogens-13-00718-t001:** A BLAST search for sequences like *L. mexicana* gp63 in different parasitic protozoa.

Description	E Value	Query Cover %	Identity %	Accession
*Leishmania mexicana* (GP63, leishmanolysin)	0.0	100	100	XP_003872886.1
*Trypanosoma cruzi* (surface protease GP63 putative)	1 × 10^−111^	86	37.86	XP_817808.1
*Entamoeba histolytica* (cell surface protease GP63 putative)	4 × 10^−37^	71	27.43	XP_652632.1
*Entamoeba dispar* SAW760 (uncharacterized protein EDI_037980)	3 × 10^−35^	81	25.81	XP_001740726.1
*Entamoeba invadens* IP1(hypothetical protein EIN_174510)	1 × 10^−31^	66	27.59	XP_004184102.1
*Acanthamoeba castellanii* str. Neff (leishmanolysin, putative)	5 × 10^−45^	52	32.75	XP_004337275.1
*Naegleria fowleri* (unnamed protein product)	3 × 10^−30^	52	28.25	XP_0044566011.1

**Table 2 pathogens-13-00718-t002:** Predicted functions of gp63 homologues in Trypanosomatids and Amoeba species.

Species	Gene Bank ID	Uniprot ID	TriTrypDB/ AmoebaDB	Name of Protein	Gene Ontology Predicted Functions
* L. mexicana *	XP_003872886.1	E9AN57	LmxM.10.0470	GP63, leishmnolysin	GO: 0004222 metalloendopeptidase activity GO: 0005886 plasma membrane GO: 0006508 proteolysis GO: 0007155 cell adhesion GO: 0016020 membrane GO: 0046872 metal ion binding
* T. cruzi *	XP_817808.1	Q4DTV2	TcCLB.508693.100	Surface protease GP63 putative	GO: 0004222 metalloendopeptidase activity GO: 0005886 plasma membrane GO: 0006508 proteolysis GO: 0007155 cell adhesion GO: 0016020 membrane GO: 0046872 metal ion binding
* E. histolytica *	XP_655632.1	C4M655	EH.042870	Cell surface protease gp63puttive	GO: 0004222 metalloendopeptidase activity GO: 0005886 plasma membrane GO: 0006508 proteolysis GO: 0007155 cell adhesion GO: 0008270 zinc ion binding membrane GO: 0016020 membrane
* E. dispar *	XP_001740726.1	B0ERK0	EDI_037980	Hypothetical protein conserved	GO: 0004222 metalloendopeptidase activity GO: 0005886 plasma membrane GO: 0006508 proteolysis GO: 0007155 cell adhesion GO: 0016020 membrane GO: 0016021 integral component of membrane
* E. invadens *	XP_004184102.1	A0A0A1TW87	EIN_174510	Hypothetical protein	GO: 0004222 metalloendopeptidase activity GO: 0006508 proteolysis GO: 0007155 cell adhesion GO: 0016020 membrane GO: 0016021 integral component of membrane
* A. castellanii *	XP_004337275.1	L8GQS8	ACA1_29880	Leishmanolysin putative	GO: 0004222 metalloendopeptidase activity GO: 0006508 proteolysis GO: 0007155 cell adhesion GO: 0016020 membrane
* N. fowleri *	XP_044566011.1	A0A6A5C651	NF0068990	Metalloendopeptidase zinc ion binding protein	GO: 0004222 metalloendopeptidase activity GO: 0006508 proteolysis GO: 0007155 cell adhesion GO: 0016020 membrane

**Table 3 pathogens-13-00718-t003:** Values of the superposition of the structure’s analysis. (Length) The number of pairs of residues that are structurally equivalent; (QC %) Query cover based on alignment length, expressed in percent; (TC %) Target cover based on alignment length, expressed in percent; (Score) Measure of structural similarity; (RMS) Root-mean-square error of the superposition in Ångströms; and (SI %) Sequence identity of the query and target in the equivalent regions, expressed in percent.

Description	Length	QC%	TC%	Score	RMS	SI%
*Leishmania mexicana*	465	100	98	465	0.07	81
*Trypanosoma cruzi*	444	95	97	442	0.44	39
*Entamoeba histolytica*	424	91	89	420	0.73	25
*Entamoeba dispar*	405	87	89	400	0.79	26
*Entamoeba invadens*	418	89	93	414	0.69	20
*Acanthamoeba castellanii*	377	81	91	373	0.73	31
*Naegleria fowleri*	432	93	90	428	0.68	28

**Table 4 pathogens-13-00718-t004:** Binding energies and molecular interactions of AA with parasite targets. Catalytic site residues are highlighted in blue for glutamate and in red for histidine. Residues shown in black represent amino acids that interacted with AA within a 5 Å radius.

Organism	Interaction Residues on the Binding Site	H-Bond	Energy (kcal/mol)
*L. mexicana*	VAL223 SER222 ALA348 SER330 PRO347 ALA346 HIS334 SER333 SER273 HIS268 GLU265 TRP226 HIS264	HIS264	−11.16
* T. cruzi *	GLU452 GLY453 LEU367 LEU368 TYR379 ARG416 PRO417 VAL419 TYR418	TYR379 ARG416	−8.84
*E. histolytica*	ARG261 TYR161 TYR126 HIS210 GLU207 HIS206 THR282 ILE281 GLY280 HIS267 ALA266 THR263 GLY262 ASP215	HIS206	−9.57
*E. dispar*	SER155 SER157 GLU392 TYR395 THR282 SER283 ILE281 GLY262 THR263 GLY280 ALA266 HIS267 HIS210 HIS206 GLU207 ALA162 ALA160 TYR161 PHE159 VAL158	HIS267 GLU207	−10.32
*E. invadens*	LEU156 GLY157 PHE160 SER60 ILE232 VAL243 TYR64 ASP265 LYS440 PRO439 SER408 CYS443 ASN406 THR446 ARG153 PRO241	---	−8.92
*A. castellanii*	GLU184 LEU417 ASN420 VAL298 VAL300 GLY279 GLY297 HIS284 HIS231 HIS235 GLU232 GLY189 ALA228 ASN188 ALA187 ILE186 GLU185	---	−11.02
*N. fowleri*	GLY310 LEU312 GLU345 SER428 LYS431 LEU429 GLY430 GLY409 LEU434 GKY427 HIS414 SER413 HIS356 GLN204 HIS352 GLU353 THR200 GLY313 ASN199 VAL311	---	−10.81

## Data Availability

The original data presented in the study are openly available in NCBI [https://blast.ncbi.nlm.nih.gov/Blast.cgi]: *Leishmania mexicana* GP63, leishmanolysin (Accession No. XP_003872886.1); *Trypanosoma cruzi* surface protease GP63 putative (Accession No. XP_817808.1); *Entamoeba histolytica* cell surface protease GP63 putative (Accession No. XP_652632.1); *Entamoeba dispar* SAW760 uncharacterized protein EDI_037980 (Accession No. XP_001740726.1); *Entamoeba invadens* IP1 hypothetical protein EIN_174510 (Accession No. XP_004184102.1); *Acanthamoeba castellanii* str. Neff leishmanolysin, putative (Accession No. XP_004337275.1); *Naegleria fowleri* unnamed protein product (Accession No. XP_0044566011.1).

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
