# Peer review of "Exploration of the Binding Site of Arachidonic Acid in gp63 of *Leishmania mexicana* and in Orthologous Proteins in Clinically Important Parasites"

_pathogens, 2024, doi:10.3390/pathogens13090718_

Round 1
Reviewer 1 Report
Comments and Suggestions for Authors
Dear Authors,
The manuscript is very interesting as an extension of at least two recent Author’s published articles. Furthermore, it holds promise for future research in protozoan virulence factors, including drug discovery and immunotherapeutic product fields. Thus, scientific readers and the community will benefit.
However, a minor revision is required as follows:
Materials and methods
Line 145 – Please, check the and rectify Trypanosoma cruzi parasite evolutive form. I believe it is epimastigote or Trypomastigote forms.
Results
Figure 2: May the authors consider adding more information to the subtitle of Figure 2. For example, what means LmAXA?
Discussion
Line 641: Editorial check for the reference [Hernández-Ramírez et al., 2023].
Supplementary materials
Please, check the subtitle of Figure 1, must be improved. Is confused and should be following the main manuscript.
Overall, I have one question:
Line 147. The authors reported during the methodology the soluble fraction obtained from various parasites. Did you perform the COX activity for Trypanosoma cruzi using the COX activity kit?
Kind Regards

Reviewer 2 Report
Comments and Suggestions for Authors
The manuscript by Hernández-Ramírez and co-authors presents evidence for the ortholog of the L. mexicana gp63 protein in other protists. Although the evidence is clear, please pay attention to my comments on data interpretation, especially the BLAST and RNAseq analyses in the "Results" section, as your presentation of the results confuses me. I list the proposals below. Presented data also linked the cyclooxygenase activity to the gp63 protein in L. mexicana and the overall results represent good material for further studies.
In general, I think the work is interesting and have some suggestions below that may improve the manuscript. However, several comments need to be resolved before accepting the manuscript.
Comments:
1. I recommend simplifying the title of the article. Also, the abbreviation "COX" used for cyclooxygenase activity can be misleading because it is also used for cytochrome c oxidase activity. Therefore, I suggest using the full name of the activity in the title.
2. Materials and Methods, 2.3 Parasites: The cultivation conditions of all parasites should be listed in one section, at least in references. Section 2.3 Parasites includes only the cultivation of L. mexicana, E. histolytica, while the cultivation of donated parasites is described elsewhere in the text or nowhere.
3. Consolidate references to images in the text as required by the journal. Line 254: (Figure 1A) vs. Line 256: (Fig. 1A). Both ways of referring to figures appear in the text.
4. In Table 1, the column names are swapped, the second is e-value and the third is query coverage. E-value is not given in percentage as shown in the table.
5. Table 1: For completeness, the authors should also include the e-values for reverse hits to the organism from which they had the query at the beginning. Thereby showing that the proteins from different organisms are reciprocal best blast hits and therefore orthologs. I suggest including this information in the Supplementary material.
6. The statement “Figure 2 shows that except for A. castellanii, it was not possible to detect the existence of mRNA for the putative leishmanolysin proteins identified by BLAST” is incorrect. The results in Figure 2 show the exact opposite. Moreover, all e-values from Table 1 are low enough to claim that proteins are an ortholog of leishmanolysin. Thus, based on the BLAST and RNAseq the evidence for GP63 protein in studied organisms is convincing, apart from A. castellanii for which no mRNA was detected.
7. Authors previously showed that e D12 mAb recognizes epitopes on A. castellanii and N. fowlleri trophozoites. How do the authors explain that the antibodies recognize the protein in A. castellanii for which they did not detect the mRNA? The authors mention in the discussion that they detected the protein in A. castellanii, but the controversy with the missing mRNA in this study should be discussed in more depth.
Minor comments:
Line 54: Even a sentence starting with a Greek letter should not start with a small letter. Write "α" as “Alpha” or rephrase the sentence so that the Greek writing does not appear at the beginning.
Line 56: “PGE2” is obviously a designation for prostaglandin, but for easier reading the abbreviation should be introduced, or even not be used if the term does not appear further in the text.
Line 106: “in LB with ampicillin” in LB medium with ampicillin. LB stands for lysogenic broth, so I missed that the bacteria were grown in the medium.
Line 121: Missing composition of TBST buffer.
Line 130: Centrifugation conditions expressed as acceleration "g" should not contain an "x", replace 1000xg with the correct form 1000g, and likewise in any other centrifugation description.
Line 172: The composition of encystation medium is missing.
Line 249: “CIcloxygenase” Is this correct?
Line 325: Redundant full stop after the sentence.
Line 385: Redundant full stop after the sentence.
Line 456: Redundant full stop after the sentence.
Line 497: Redundant full stop after the sentence.
Line 519: Redundant full stop after the sentence.
Line 567: Remove the full stop from the abbreviation for arachidonic acid (A.A to AA).
Figure 8: Unify the use of closures in the legend, it is either one A), or both (C).
Line 573: Redundant full stop after the sentence.
